Physiological response and transcriptome analyses of leguminous Indigofera bungeana Walp. to drought stress

Ma Shuang 1
Hu Haiying haiying@nxu.edu.cn 1 2
Zhang Hao 1
Ma Fenghua 1
Gao Zhihao 1
Li Xueying 1
1 College of Forestry and Prataculture, Ningxia University , Yinchuan , Ningxia , China
2 Breeding Base for State Key Laboratory of Land Degradation and Ecological Restoration of North-Western China, Ningxia University , Yinchuan , Ningxia , China
Abd El-Moneim Diaa
Electronic publication date: 2023 Jun 14
Publication date: 2023
Volume: 11
Electronic Location ID: e15440
Received 2022 Nov 28; Accepted 2023 Apr 28
Copyright: ©2023 Ma et al.
Copyright year: 2023
Copyright holder: Ma et al.
License: This is an open access article distributed under the terms of the Creative Commons Attribution License, which permits unrestricted use, distribution, reproduction and adaptation in any medium and for any purpose provided that it is properly attributed. For attribution, the original author(s), title, publication source (PeerJ) and either DOI or URL of the article must be cited.
License URL: https://creativecommons.org/licenses/by/4.0/

Keywords: Drought stress, Differential gene expression, Regulatory pathways, Transcriptomics, Indigofera bungeana Walp

Funding: The National Natural Science Foundation of China 32160406 The First-class Discipline Construction (Grassland Science Discipline) for the high school in Ningxia NXYLXK2017A01 Natural Science Foundation of Ningxia Province 2022AAC03080 This study was funded by the National Natural Science Foundation of China (NO.32160406), the First-class Discipline Construction (Grassland Science Discipline) for the high school in Ningxia (NO. NXYLXK2017A01), and the Natural Science Foundation of Ningxia Province (NO.2022AAC03080). The funders had no role in study design, data collection and analysis, decision to publish, or preparation of the manuscript.

==============================
Objective

Indigofera bungeana is a shrub with high quality protein that has been widely utilized for forage grass in the semi-arid regions of China. This study aimed to enrich the currently available knowledge and clarify the detailed drought stress regulatory mechanisms in I. bungeana, and provide a theoretical foundation for the cultivation and resistance breeding of forage crops.

Methods

This study evaluates the response mechanism to drought stress by exploiting multiple parameters and transcriptomic analyses of a 1-year-old seedlings of I. bungeana in a pot experiment.

Results

Drought stress significantly caused physiological changes in I. bungeana. The antioxidant enzyme activities and osmoregulation substance content of I. bungeana showed an increase under drought. Moreover, 3,978 and 6,923 differentially expressed genes were approved by transcriptome in leaves and roots. The transcription factors, hormone signal transduction, carbohydrate metabolism of regulatory network were observed to have increased. In both tissues, genes related to plant hormone signaling transduction pathway might play a more pivotal role in drought tolerance. Transcription factors families like basic helix-loop-helix (bHLH), vian myeloblastosis viral oncogene homolog (MYB), basic leucine zipper (bZIP) and the metabolic pathway related-genes like serine/threonine-phosphatase 2C (PP2C), SNF1-related protein kinase 2 (SnRK2), indole-3-acetic acid (IAA), auxin (AUX28), small auxin up-regulated rna (SAUR), sucrose synthase (SUS), sucrosecarriers (SUC) were highlighted for future research about drought stress resistance in Indigofera bungeana.

Conclusion

Our study posited I. bungeana mainly participate in various physiological and metabolic activities to response severe drought stress, by regulating the expression of the related genes in hormone signal transduction. These findings, which may be valuable for drought resistance breeding, and to clarify the drought stress regulatory mechanisms of I. bungeana and other plants.

Introduction

Drought is typical adverse stress during plant lifecycle that inhibits the plant growth, development, and yield (Martin-StPaul, Delzon & Cochard, 2017). Drought stress impacts plant growth and development, morphological structure, reactive oxygen (ROS) metabolism, osmoregulation, signal transduction, gene expression regulation, secondary metabolites, membrane transport, and energy metabolism (Bashir, Matsui & Rasheed, 2019). Plants may alter their morphology in response to drought stress to reduce water loss and increase water utilization (Kaur et al., 2021). Plant roots can detect water deficit signals and perform stress transduction to stimulate the expression of genes and transcription factors to coordinate various physiological and metabolic activities, regulate cellular osmotic potential, reduce water loss, protect cellular membrane systems, and maintain normal cellular physiological processes (Chinnusamy, Schumaker & Zhu, 2004).

Recently, many genes and metabolic pathways in response to drought stress have been uncovered (Mathur & Roy, 2021). Additional research has been conducted on transcription factors. The main transcription factors associated with plant stress resistance are avian myeloblastosis viral oncogene homolog (MYB), bHLH, WRKY, bZIP, AP2/EREBP, among others. A meta-analysis study on transcription factor enrichment in Arabidopsis under drought stress found that WRKY, AP2/ERF, bHLH, MYB, and bZIP transcription factor families as the most enriched, 56% of common genes were regulated by these transcription factor families under drought stress (Sharma et al., 2018). A total of 301 AP2/ERF transcription factor families were identified in soybean under drought conditions. Relative to the wild-type, transgenic plants overexpressing the GmAP2/ERF144 gene in soybean considerably lower the electrical conductivity and MDA content (Wang et al., 2022). Yoshida et al. (2010) found that, under drought stress, the bZIP family ABF3, associated with the ABA hormone and stimulated the activation of genes related to water stress downstream of drought stress such as late embryogenesis abundant, PP2C etc. In response to drought stress, genes associated with abscisic acid (ABA) metabolism, root elongation, and peroxidase activity were dramatically up-regulated in alfalfa, Ss, and lignin levels and were also significantly associated with drought resistance (Ma et al., 2021).

I. bungeana is a small shrub of the genus Indigofera L. in the Leguminosae family, which is widely distributed in South and East China, as well as Shanxi and Hubei in China, with a resource characteristic of strong ecological adaptability, high forage value, and strong soil and water conservation capacity (Zheng et al., 2011). This study has shown that the crude protein content of I. bungeana was two times that of crude fiber, which met the requirements of high quality protein (Xu et al., 2017). The shrub has resilient characteristics adaptable to ecological restoration, and its water use efficiency is significantly higher than that of Lespedeza bicolor and Amorpha fruticosa, displaying greater drought resistance (Ran et al., 2019). Yu et al. (2008) measured the physiological indexes such as superoxide dismutase, proline and malondialdehyde through natural drought I. bungeana, and the results showed that I. bungeana has comprehensive advantages in drought resistance. However, there is scarcity of information from empirical research on the physiological and molecular mechanisms of the drought resistance of I. bungeana is limited.

Therefore, this study aimed to investigate the physiological and molecular mechanisms of adaptability to drought stress as well as gene-mining for drought-tolerance genes of I. bungeana. Hence, we analyzed the response mechanisms to water stress by exploiting multiple parameters and transcriptomic analyses of a 1-year-old seedlings of I. bungeana in a pot experiment.

Materials & Methods

Test materials

The seeds of the domesticated cultivar “Wushan I. bungeana” were obtained from by Fan Yan, Researcher, Chongqing Academy of Animal Husbandry.

Experimental treatment

The experiment was conducted from June to August, 2021 at the grass science laboratory of Ningxia University and the glass-greenhouse of the Agricultural Experimental Practice Base of the same university. Plastic pots with a diameter of 20 cm and a height of 20 cm were used for the experiment. Each pot contains 2.5 kg of sandy loam soil with a field water holding capacity of 21.56% and a soil capacity of 1.52 g cm−3 with a saturation water holding capacity of 29.60%. When the seedlings reached over five cm in height and developed a healthy root system, they were transplanted into cultivation containers. To limit water evaporation, the soil was coated with polyethylene plastic granules. Natural light was used and the temperature in the glass-greenhouse was 28 °C/18 °C (day/night).

Two water treatments: With normal water supply as control (70–80% of the field’s water content), and using potted natural drought as drought (20–30% of the field’s water content) were used for the experiment. Each basin had three replications, there are 30 pots. To grasp the replenishment pattern, soil moisture content was measured by weighing at a fixed time in the afternoon every day, whereas volumetric soil moisture content was measured using a portable TDR (Mini Trase with soil-moisture; TDR Technology, Monroe, NY, USA), and the amount of water required to replenish the soil was calculated to reach a set interval. The control group poured 2 L of water every 3 d, and the treatment group conducted continuous drought stress until the soil water content was about 20% of the field water quantity. After four weeks, the content of water in the 2.5 kg soil was reduced from 2 L to about 500 ml. After four weeks of water control treatment, the sampling and testing began. Each sample was conducted three replications, and all the leaf and root samples were snap-frozen in liquid nitrogen and stored at −80 °C for enzyme activity analysis and RNA-sequencing.

Measurement of indicators and methods

Determination of biomass characteristics and root distribution

The leaves, stems, and roots were separated; the surface water was blotted out with filter paper before weighing; the fresh weight of the leaves, stems, and underground roots were weighed separately on a 1/1000 balance. The samples were then placed in an oven at 105 °C for 30 min, and subsequently dried at 70 °C to a constant weight, and the dry weight of each part was determined after cooling.

To evaluate the root distribution features, five plants for each treatment were rinsed with deionized water, and then placed in a transparent tray filled with 10–15 mL distilled water, and sorted. The plants were scanned using a root scanner (EPSON expression) with a resolution of 300 dpi, and WinRHIZO root analysis system software was used to analyze the root images to determine the root length, root volume, root surface area, and the number of root branches (Li, 2000).

Photosynthetic gas exchange parameters

At 9:00–11:00 am, five fully expanded leaves of the same size and orientation were selected from each treatment and replicated three times for the analysis of photosynthetic parameters. Pn, Gs and Tr were measured using Li-6400 portable photosynthesizer (LI-COR, USA). At 25 °C, flow value of 500 µmol s−1, carbon dioxide value of 400 µmol mol−1, and light intensity of 1,000 µmol mol−2 s−1 were recorded.

Measurement of various physiological parameters

Weighing method was used for measuring leaf relative water content. The dried biomass samples (leaf, stem, and root) were pulverized and sieved through a 100-mesh sieve before sending the samples to the Huake Precision Stable Isotope Laboratory for the analyses of δ13C values for each organ. CT was determined by direct extraction method (Sun et al., 2022). The activity of CAT was measured by UV spectrophotometry kit (Li et al., 2021a; Li et al., 2021b), and the content of H2O2 was measured by visible light spectrophotometry (Yamazaki, 1967). Thiobarbituric acid method was used to determine the content of MDA (Yang et al., 2020), and acidic ninhydrin spectrophotometer method was used to determine the content of Pro (Chen & Wang, 2006).

Transcriptome sequencing analysis

A total of 12 samples inclued two tissues in leaf and root of seedling and two treatments (control and drought) and three biological replications were used for transcriptome analysis. A total amount of 1 µg RNA per sample was used as input material for the RNA sample preparations. Sequencing libraries were generated using NEBNext®Ultra™ RNA Library Prep Kit for Illumina® (NEB, Ipswich, MA, USA) following manufacturer’s recommendations and index codes were added to attribute sequences to each sample. Then PCR was performed with Phusion High-Fidelity DNA polymerase, Universal PCR primers and Index (X) Primer. At last, PCR products were purified (AMPure XP system) and library quality was assessed on the Agilent Bioanalyzer 2100 system. The library preparations were sequenced on an Illumina Hiseq 2000 platform and paired-end reads were generated (Grabherr, Haas & Yassour, 2011).

The sequences were further processed with a bioinformatic pipeline tool, BMKCloud (http://www.biocloud.net) online platform. The quantity and quality of RNA were assessed by measuring Illumina (NEB, Ipswich, MA, USA). After quality control, raw data was collected, and Q20, Q30, GC-content, and sequence repeat levels were calculated for clean data. All subsequent analyses were built on high-quality clean data. In other words, after filtering clean data (reads), a database comparison was performed to obtain data for subsequent transcript assembly, expression calculation, and so on, as well as a quality assessment of the RNA-seq comparison results. The cloud platform developed by QingdaoBemac Biotechnology Co. was used to analyze transcriptome data. The statistical power of this experimental design of calculated in RNASeqPower is 0.92, 0.99. Computing the power values using RNA-Seq (https://rodrigo-arcoverde.shinyapps.io/rnaseq_power_calc).

Gene function annotation

We used DIAMOND (version:v2.0.4) (Buchfink, Xie & Huson, 2015) to align the Unigene sequence to the NR, Swiss-Prot, COG, KOG, eggNOG4.5, and KEGG databases by selecting the BLAST parameter E-value not greater than 1e−5 and the HMMER parameter E-value not greater than 1e−10. Gene functions were annotated based on the following databases: NR (NCBI non-redundant protein sequences); Pfam (protein families); KOG/COG/eggNOG (protein immediate homology groups); Swiss-Prot (manually annotated and reviewed protein) sequence database); KEGG (Kyoto Encyclopedia of Genes and Genomes); GO (Gene Ontology).

KEGG pathway enrichment analysis

KEGG (Kanehisa et al., 2004) is a database resource for understanding high-level functions and utilities of the biological system, such as the cell, the organism and the ecosystem, from molecular-level information, especially large-scale molecular datasets generated by genome sequencing and other high-throughput experimental technologies (http://www.genome.jp/kegg/). We used KOBAS (Xie et al., 2011) to test the statistical enrichment of differential expression genes in KEGG pathways.

Differential gene screening

DEGs were identified by comparing sample controls with treatment groups, and DESeq2 provides statistical routines for determining differential expression in numerical gene expression data using a model based on a negative binomial distribution. The resulting P-values were adjusted using the Benjamini and Hochberg method to account for the false discovery rate (FDR). DESeq2 identified genes with adjusted P-values (0.05) were considered differentially expressed. P-values were adjusted using q-values. The screening criteria for genes significantly differentially expressed were q-value 0.01 and — log2 (fold change) —>1.5.

Real-time fluorescence quantitative PCR (qRT-PCR) validation

qRT-PCR quantification experiments were conducted using kits (Shanghai Yisheng Biological Co., Shanghai, China) Nine DEGs were randomly selected for qRT-PCR quantification (PCR instrument AB-7500), BMK_UniGene_169182 for leaves, and BMK_UniGene_279437 for roots (Table 1), and finally, the expression was analyzed using the 2−ΔΔCt (Livak & Schmittgen, 2001).

Table 1 Primer sequences of differential genes.

Organ	Gene name	Primer	Sequence(5′–3′)	
MJ-LEAF	BMK_UniGene_038077	Forward primer	CCACCATAGAACCAGACCATAG	
Reverse primer	GTGGACGCTGACAAGGATAA	
BMK_UniGene-176857	Forward primer	ACAATGTAGGCAAGGGAAGT	
Reverse primer	GAGCAGTCCACCGACAG	
BMK_UniGene-005090	Forward primer	AGTGACTCTGACATCCCATAC	
Reverse primer	GGAAATTGGCTGCTGTGAAA	
BMK_UniGene-121193	Forward primer	CAACTACTGAACCTAGCCACTAC	
Reverse primer	TCCTTTGATTGCTCAGACTACTT	
BMK_UniGene_169182	Forward primer	AGTGGTCGTACAACTGGTATTG	
Reverse primer	AGCATGTGGGAGAGCATAAC	
MJ-ROOT	BMK_UniGene-122825	Forward primer	CGAACCACGAAGTGCAAATAG	
Reverse primer	CTGAGATGCTGGCCATGTATAA	
BMK_UniGene-030285	Forward primer	AGGGCCGTGGAGTTTATTG	
Reverse primer	GATTCCAAGCTTCTTCGGTACT	
BMK_UniGene-029024	Forward primer	CAAGCTCCACCGAAGTAACA	
Reverse primer	CTTCTCCGTTAGCCCTTTCTT	
BMK_UniGene-129423	Forward primer	CCGAACGCTCAAACAACTATG	
Reverse primer	GAAGCACATCCCGCAAATATC	
BMK_UniGene-009176	Forward primer	TGTTCAGTTCCATACACCTTCTC	
Reverse primer	CGTCCCTCGCATTCTCATTAT	
BMK_UniGene_279437	Forward primer	GTTGAGACTTTCTCTCCGACTATCC	
Reverse primer	GGGTCTTTCTTCTCCACATTCT	

Data processing and analysis

Data were statistically processed using Microsoft WPS, GraphPad 8.0.2 for one-way ANOVA, and for graphing, The correlations between plant-related assays under drought stress in I. bungeana were assessed via Pearson’s product-moment correlation with results expressed as mean ± standard error (SE).

Results

Effect of drought stress on growth performance of I. bungeana

Table 2 revealed that drought stress inhibited the growth of I. bungeana roots, which is also illustrated in Fig. 1. Under the drought stress, total root length, root surface area, root volume, root branches, root biomass, as well as stem biomass, leaf biomass, and specific leaf area (SLA) of I. bungeana were all significantly reduced relative to CK, but the root to crown ratio(RCR) was significantly increased compared with CK (p < 0.05).

Table 2 Effects of drought stress on growth performance of I. bungeana.

Lowercase letters represent significant differences between treatments (P < 0.05). The F value is the ratio of two mean squares.

Treatment	Control	Drought	F	P	
Total root length (cm)	1,383.39 ± 89.13a	484.83 ± 30.46b	15.71	0.01	
Root surf area (cm2 )	158.54 ± 9.12a	49.31 ± 5.38b	106.39	0.00	
Root volume (cm3 )	1.48 ± 0.09a	0.40 ± 0.06b	95.07	0.00	
Forks	4,348.33 ± 75.45a	1,286.33 ± 50.73b	10.89	0.03	
SLA (cm2 g−1 )	516.58 ± 23.99a	272.28 ± 15.39b	32.64	0.00	
Root biomass (g)	4.04 ± 0.01a	3.35 ± 0.14b	72.09	0.00	
Stem biomass (g)	5.01 ± 0.20a	3.82 ± 0.19b	17.69	0.01	
Leaf biomass (g)	4.86 ± 0.18a	4.04 ± 0.11b	14.85	0.01	
RCR (g)	0.43 ± 0.10a	0.57 ± 0.11b	105.59	0.00	

Figure 1 Phenotypic responses of I. bungeana for drought stress.

The scale is 5 cm, the left plant is the control and the right plant is the drought treatment. Under drought stress, leaves turn yellow and root distribution decreases.

Effect of drought stress on physiological parameters of I. bungeana

As shown in Fig. 2, the physiological parameters of I. bungeana changed significantly under different drought degrees. The stomatal conductance (Gs) was 43% lower than that of the CK (p < 0.001) (Fig. 2A), Net photosynthetic rate (Pn) was 67% lower than that of the CK (p < 0.001) (Fig. 2B), transpiration rate (Tr) was 67% lower than that of the CK(p < 0.05) (Fig. 2C), the total chlorophyll content (CT) was 43% higher than that of the CK (p < 0.05) (Fig. 2D). Pro was 354% higher than that of the CK (p < 0.01) (Fig. 2E). Ss was 31% higher than that of the CK (p < 0.001) (Fig. 2F), MDA was 43% higher than that of the CK (p < 0.01) (Fig. 2G). CAT was 80% higher than that of the CK (p < 0.01) (Fig. 2H), H2O2 was 93% higher than that of the CK(p < 0.001) (Fig. 2I), The RWC was 21% lower than that of the CK (p < 0.01) (Fig. 2J), Sc was 20% lower than that of the CK (p < 0.05) (Fig. 2L), The δ13C values of roots, stems, and leaves significantly increased, and the δ13C values of roots and stems were significantly greater than those of leaves (p < 0.05) (Fig. 2K).

Figure 2 Changes in Gs (A), Pn (B), Tr (C), CT (D), Pro (E), Ss (F), MDA (G), CAT (H), H2O2 (I), RWC (J), δ13 C (K), Sc (L) of I. bungeana under control and drought stress.

Gs, stomatal conductance; Pn, Net photosynthetic rate; Tr, transpiration rate; CT, chlorophyll content; H2O2, hydrogen peroxide; CAT, catalase; RWC, relative water content; MDA, malondialdehyde; Pro, proline; Ss, soluble sugar; δ13C, δ13C values; Sc, starch.

Correlation analysis among the physiological parameters of I. bungeana under drought stress.

Different degrees of correlations existed between the physiological parameters (Fig. 3). MDA was positively correlated with Pro and CT (p < 0.05), while negative correlation exists between Sc, total root length, root surfarea and root volume (p < 0.05). The Ss was positively correlated with H2O2 and CAT (p < 0.05). The Sc has a positive correlation with total root length, root surfarea and root volume (p < 0.05).

Figure 3 Correlation analysis between different physiological indexes.

The horizontal and vertical coordinates are the physiological indicators, the size of the circle represents the size of the absolute value of the correlation, the scale represents turning red indicates a gradual increase in positive correlation and turning blue indicates a gradual increase in negative correlation.

Transcriptome sequencing data evaluation

Twelve cDNA libraries were created by RNA-Seq, and the raw data of each sequenced library is presented in Table 3. The RNA-Seq readable data of the 12 samples were 19777998–2383777 with Q30 values over of 94.12%. The measured data were highly accurate and of good quality and were subsequently used for data analysis.

Table 3 Sample sequencing and data alignment statistics.

BMK-ID	GC Content	% ≥Q30	Clean reads	Mapped reads	Mapped ratio	
MJWCKL1	44.11%	95.50%	21,595,620	15,272,591	70.72%	
MJWCKL2	44.33%	95.14%	23,837,776	17,089,601	71.69%	
MJWCKL3	44.47%	95.32%	21,084,164	15,135,628	71.79%	
MJWCKR1	43.53%	94.12%	22,016,955	14,998,664	68.12%	
MJWCKR2	43.22%	94.56%	21,527,882	14,878,497	69.11%	
MJWCKR3	43.57%	94.82%	21,457,173	14,808,545	69.01%	
MJWTrL1	44.04%	94.81%	22,273,782	16,048,743	72.05%	
MJWTrL2	44.38%	94.48%	21,784,723	15,671,948	71.94%	
MJWTrL3	44.05%	95.07%	20,800,640	14,808,175	71.19%	
MJWTrR1	43.99%	95.11%	19,777,998	14,029,897	70.94%	
MJWTrR2	43.61%	94.70%	21,463,868	15,003,528	69.90%	
MJWTrR3	43.54%	94.61%	21,368,807	15,049,595	70.43%	
Notes.

* BMK-ID, Biomarker sample analysis number.

GC Content, Clean Data GC content is the percentage of G and C bases in Clean Data in the total bases. % ≥ Q, The percentage of bases with a Clean Data quality value greater than or equal to 30. Clean Reads, The number of Clean Reads is double-ended. Mapped Reads, The number of Mapped Reads is doubleended; Mapped Ratio: The proportion of Mapped Reads in Clean Reads.

Screening for DEG

Drought stress gene correlation analysis revealed that 3,978 different expression genens (DEGs), including 2,934 up-regulated genes and 1,053 down-regulated genes, were identified in the leaves of I. bungeana under drought stress (Fig. 4A). A total of 6,923 DEGs, including 2,903 up-and 4,020 down-regulated genes, were identified in the roots of I. bungeana. These results suggested that roots were more sensitive drought stress than leaves. A total of 207 (2.2%) DEGs were identified opposite expression in the leaves and roots (Fig. 4C).

GO functional enrichment analysis of DEG

In order for plant leaves and roots to adapt to unfavorable enviroment, their biological process have coordinated throughout the plant level. The results presented in Fig. 5 indicates that the DEGs in the leaves and roots are significantly enriched in seventeen biological process categories, fifteen cellular component categories, and eleven molecular functions categories. The roots enriched with more DEGs than thoes from leaves, the “metabolic” was most siginificantly enriched in leaves and roots tissues, follow by “cellular” and “catalytic activity”.

Figure 4 (A) Statistical diagrams of the number of DEGs (P< 0.05, log2 FC ≥ 2). (B) Venn diagrams of the number of DEGs. (C) Venn diagrams of the number of DEGs.

a, Leaf-up; b, Leaf-down; c, Roots-up; d, Roots-down; G0, Leaf; G1, Roots.

Figure 5 GO functional enrichment of control and treatment DEGs in I. bungeana, leaf (A) and root (B).

The horizontal axis is the GO classification, the left side of the vertical axis is the percentage of the number of genes, and the right side is the number of genes. This figure displays the gene enrichment of each secondary function of GO under the background of differentially expressed genes and the background of all genes, reflecting the status of each secondary function in the two backgrounds, and secondary functions with significant proportion differences indicate differentially expressed genes.

Transcription factor analysis

As illustrated in Fig. 6, 150 transcription factors were identified in the DEGs of the leaves, divided into 19 families. The top four DEGs in numerical order were 16 in the bHLH family, 11 in the MYB-related family, ten in thep-coumarate 3-hydroxylase (C3H) family, nine in the Plant Homeodomain Finger (PHD) family. A total of 304 transcription factors were identified in the DEGs of the roots, which were divided into 19 families. The top four in quantitative order were 36 in the AP2/ERF-ERF family, 33 in the bHLH family, 25 in the Cys2/His2 (C2H2) family, 24 in the bZIP family. The most represented transcription factors commonly expressed in leaves and roots were bHLH, MYB bZIP. A total of 49 bHLH, 33 MYB and 38 bZIP transcription factor family genes were found to be abundant in the root and leaf tissues.

Figure 6 Differential transcription factors of I. bungeana in response to drought stress: (A) Leaf, (B) root.

The ordinate represents the type of transcription factor, and the abscissa represents the number of transcription factors.

DEGs KEGG metabolic pathway enrichment analysis

To elucidate the functional enrichment and metabolic pathways of DEGs under drought stress, Kyoto Encyclopedia of Genes and Genomes (KEGG) metabolic pathway enrichment analysis was performed on DEGs in response to drought stress. The top 20 pathways of leaves and roots were screened as the strong response pathway. As illustrated in Figs. 7A & 7C, “plant hormone signaling transduction”, “starch and sucrose metabolism” were the most intensive physiology activities in the leaves. “glycolysis/gluconeogenesis”, “protein processing in the endoplasmic reticulum”, “MAPK signal pathway-plant”, “plant hormone signaling transduction” were the most active activeties in the roots (Figs. 7B & 7D). “plant hormone signaling transduction” was the most enriched pathway in the leaves and roots. There were 81 up-regulated and 34 down-regulated DEGs in leaves. There were 133 down-regulated DEGs were significantly enriched were plant hormone signaling transduction in roots responsing to drought stress.

Figure 7 Scatterplot of enriched KEGG pathways for DEGs under drought stress. Only the top 20 most strongly of I. bungeana leaf and root represented pathways are displayed.

Each circle/triangle in the figure represents a KEGG pathway, the ordinate represents the name of the pathway, and the abscissa is the enrichment factor. The size of the circle represents the number of genes enriched in the pathway. The larger the number, the more genes.The larger the enrichment factor, the more significant the enrichment level of differentially expressed genes in this pathway. The color of the circle/triangle represents the q value, which is the P-value after correction for multiple hypothesis testing. The smaller the q value, the more reliable the enrichment significance of the differentially expressed genes in the pathway.

Analysis of DEGs associated with key metabolic pathways

We further compared four enriched KEGG pathways between the leaves and roots: “plant hormone signaling transduction”, “MAPK signal pathway-plant”, “starch and sucrose metabolism”, and “glycolysis/gluconeogenesis”. After drought stress, A total of 115 DEGs were found in the leaves, whereas 206 were found in the roots. Several plant hormone signal-related DEGs were enriched in the leaves and roots, such as ABA, IAA, ethylene (ET). Figure 8 depicted the pathway of plant hormone signaling transduction, the most DEGs were involved in the ABA, Auxin. Thirteen PP2C genes were upregulated in leaves, two SNF1-related protein kinase 2(SnRK2) were downregulated in roots to response the ABA accumulation, eight ABF genes were downregulated in leaves. While most of the Auxin metabolism-related genes were downregulated in both tissues and involved twelve SAUR-related genes and eleven GH3-related genes, nine ADP-ribosylation factor (ARF) were down-regulated in roots, and eight ARF genes were down-regulated in leaves.

Figure 8 Heatmap expression profile of DEGs associated with plant hormone signaling transduction.

The color scale indicates gene expression numbers. Blue indicates upregulated genes, and red indicates downregulated genes. The horizontal axis of the heatmap represents upregulated in leaves; downregulated in leaves; upregulated in roots and downregulated in roots from left to right.

The MAPK signaling pathway is associated with disease resistance and ROS scavenging. In the roots, 206 DEGs were found to be considerably enriched. Figure 9 demonstrated that two DEGs were down-regulated to SnRK2 in roots, thirteen DEGs were up-regulated to PP2C in leaves. The ROS-related genes of seven respiratory burst oxidase (RbohD) genes and five serine/threonine-protein kinase (OXI1) were enriched in roots. Two CAT1 were up-regulated in leaves in the drought stress pathway. Among ER/ERLs stimulated by ERF1/2, six DEGs were down-regulated to ERF1/2 in leaves and roots.

Figure 9 Heatmap expression profile of DEGs associated with MAPK signal pathway-plant.

The color scale is the same as in Fig. 8.

According to KEGG pathway enrichment, the up-regulated DEGs in the roots were mostly enriched in the glycolysis/gluconeogenesis pathway. Figure 10 demonstrated that the up-regulated DEGs in the leaf were significantly enriched in the starch and sucrose metabolism pathway, because five DEGs were up-regulated to sucrose synthase (SUS), one DEG was up-regulated to beta-fructofuranosidase (INV), three DEGs were up-regulated to beta-glucosidase, and nine DEGs were up-regulated to alpha-amylase (α-amylase). In the roots, four DEGs were up-regulated to SUS, three DEGs were up-regulated to beta-glucosidase, and four DEGs were up-regulated to beta-amylase. Aldose 1-epimerase (GALM) and pyruvate decarboxylase (PDC) were up-regulated in the glycolysis/gluconeogenesis pathway in the roots.

Figure 10 Starch and sucrose metabolism and glycolysis/glyco-isomerization pathways.

The horizontal axis of the heatmap represents upregulated in leaves, and upregulated in roots from left to right. The color scale indicates gene expression numbers, turning from blue to red indicates a gradual decrease in the number.

From Fig. 11 shows that DELLA protein, AUX 28, IAA, SAUR in the plant hormone signaling transduction were only up-regulated in leaves. SUS and beta-glucosidase were oppositely expressed in leaves and roots.

Figure 11 Heatmap of oppositely expressed genes in leaves and roots of I. bungeana on significantly enriched pathways.

The horizontal axis of the heatmap represents upregulated in leaves, and upregulated in roots from left to right. The color scale indicates log2 (fold change).

qRT-PCR validation

The expression levels of nine randomly selected DEGs of I. bungeana under drought stress was determined using the 2−ΔΔCT method. The linear regression analysis of RNA-seq and qRT-PCR expression patterns revealed that RNA-seq and qRT-PCR expression patterns were strongly associated, with a correlation coefficient of 0.78 (Fig. 12A). Figure 12B demonstrates that, except for BMK unigene038077 (caffeoyl-CoA O-methyltransferase) and BMK unigene009176 (SUS), the other gene expression patterns were consistent with the RNA-seq results.These findings indicated that the RNA-seq data agree with the expression patterns discovered by qRT-PCR analysis and can be used for further investigation.

Figure 12 (A) Correlation analysis of screening differential gene RNA-seq and qRT-PCR. (B) Histogram analysis of screening differential gene RNA-seq versus qRT-PCR.

Discussion

Plants develop various mechanisms to withstand adversities from different form of stress including drought stress. The various methods adopt by plants to withstand drought stress include morphological changes, stomatal closure regulation, osmotic substances adjustment, signal transduction, gene expression regulation, and secondary metabolite creation (Dong et al., 2019). Owing to its root system, the plants first perceive water deficit and affect the adjustments and changes on the aboveground parts. Plants rationally allocate water received to the root system and above-ground organs to modulate biomass tolerance to drought to improve water use efficiency (Meng, 2018). Drought reduces above-ground biomass growth, reduces stomatal conductance, increases chlorophyll content, and increases intracellular ROS content to enhance the antioxidant capacity (Zhang et al., 2019). Another through osmotic regulation stabilizes membranes. Pro and Ss in the plant cells are dramatically increased due to drought stress, whereas the decrease in starch content have been associated with drought-tolerance (Du et al., 2020). Consistent with the results of this study, the physiological effects of drought stress on plants have a relatively consistent response. I. bungeana responded to drought stress by increasing water use efficiency (WUE) and coordinating related physiological and metabolic functions to maintain a constant water balance. However, drought stress dramatically lowered the root distribution structure and biomass, increased the root-shoot ratio, limited photosynthesis, and regulated osmotic substances in I. bungeana. We subsequently investigated many critical genes are identified to have a function in stomatal closure on I. bungeana responds to drought stres.

A total of 10,901 DEGs were identified in I. bungeana under drought stress through transcriptome sequencing analysis, the roots had 2,945 DEGs more than the leaves(Fig. 4A), and only 947 were shared, and only 207 DEGs shown an opposite expression in both tissues (Fig. 4C), indicating that the mechanism of drought tolerance in I. bungeana is through a complex regulatory network controlled by polygenes.

As we all kown, TFs act as major modulators of many stress-responsive genes. Transcription factors, also known as trans-acting factors, bind to the cis-acting elements of the corresponding promoters or interact with the functional sections of other transcription factors, regulating responsive gene activation or inhibiting transcriptional expression. Several TF families have now been well identified, including bZIP (mainly AREB/ABF), AP2/ERF, NAC, bHLH, WRKY, and MYB, which are key regulaters involved in drought stresses (Manna et al., 2021). In this study, the most represented transcription factors commonly expressed in leaves and roots were bHLH, bZIP, MYB (Fig. 6). A total of 49 bHLH transcription factor family genes were found to be abundant in the root and leaf tissues (Fig. 6). Studies have demonstrated that a considerable proportion of the bHLH transcription factor family genes reacted to abscisic acid (ABA) signaling, overexpression of AhbHLH112 in peanut improved ABA levels under drought stress, increased the expression of ABA biosynthesis and response to ABA response-related genes (Li et al., 2021a; Li et al., 2021b; Hao et al., 2021), implying that ABA signaling may play a major role in drought stress response. The bHLH transcription factor family genes were needed to primarily coordinate plant growth and development and contribute to drought stress response through the regulation of stomatal development and abscisic acid production.

The MYB transcription factor family on the one hand tolerates drought stress by contributing to the regulation of stomatal movement, on the other hand contributing to the production of lignin (Li et al., 2019). Studies have demonstrated that AtMYB20, AtMYB42, and AtMYB43 regulate lignin and phenylalanine biosynthesis by activating genes that promote secondary wall development in Arabidopsis (Geng et al., 2020). Caffeoyl coenzyme A 3-O-methyltransferases (CCoAOMT) play an essential role in lignin production. Cenchrus ciliaris L. CCoAOMT responses to drought stress via modulating cell wall lignification as well as ABA and ROS signaling pathways (Chun et al., 2021). In this work, 31 MYB family genes were highly enriched in the phenylpropane biosynthesis metabolism pathway in the roots of I. bungeana. Among them, ABF is a class of alkaline leucine zip proteins specifically reacting to an essential regulator (ABRE; ABA-responsive element) in the ABA response pathway and belongs to the a subfamily of the bZIP family that inhibit photosynthesis through stomatal control (Wang et al., 2021). According to Kerr et al. (2018), both Arabidopsis AtABF3 and cotton GhABF2D transcription factors overexpressed in cotton cause stomatal closure and dramatically enhance drought tolerance in transgenic plants. In this study, total of 49 bHLH, 33MYB and 38 bZIP transcription factor family genes were found to be abundant in the root and leaf tissues (Fig. 6). This indicated that the bHLH, MYB, bZIP family genes may be mainly by affecting the stomatal movement, by improving the water usage efficiency (Fig. 2K) of I. bungeana in responses to water stress.

We further investigated transcriptome analysied the metabolic and biological processes of leaves and roots tissues under drought stress. The results of the KEGG pathway annotation were compatible with the GO enrichment analysied in this study. GO enrichment results shown that DEGs were mainly enriched in metabolic, and our results agreed with previous studies. Markedly, “metabolism” was the most enriched in both tissues. “Plant hormone signaling transduction” was the most enriched pathway in the leaves and roots tissues (Fig. 7). The plant hormone signal transduction pathway was considerably enriched in up-and down-regulated genes in the leaves (Fig. 7A & Fig. 7C), as well as down-regulated genes in the roots (Fig. 7D). Plant hormones are important for the regulating plant growth and development. Among them, ABA is a phytohormone that affects stomatal closure to maintain intracellular water balance and is one of the most important phytohormones in plant drought stress response (Morgil et al., 2019). DELLA proteins act as positive regulators of stomatal closure in tomato, in addition their action is enhanced by the hormone ABA, which is itself important in mediating drought stress tolerance (Sukiran, Steel & Knight, 2020). Therefore, illustrating ABA-mediated metabolic pathways may adapt to drought stress by influencing stomatal mobility in I. bungeana.

Previous studies have shown the PYR/PYL, PP2C and SnRK2 interaction with ABA and PP2C genes negatively regulate ABA responses. As a result of this interaction, SnRK2 was autophosphorylated, and ABA response element-binding factors (ABFs) were activated (Devireddy et al., 2021). The PYR/PYL protien plays a role in sensing and signaling ABA, MdPP2C24/37 from apple transgenic lines showed inhibited ABA-mediated stomatal closure, thus led to higher water loss rates (Liu et al., 2022a; Liu et al., 2022b). In this study, thirteen PP2C genes were upregulated in leaves, eight ABF DEGs were downregulated in the leaves, two SNF1-related protein kinase 2 (SnRK2) were downregulated in roots to response the ABA accumulation. Among DELLA proteins, opposite expression was seen in the leaves and roots, and it was upregulated in leaves (Fig. 11) which agrees with the results obtained from Min et al. (2020) under drought stress. This is consistent with the results of physiological studies, that drought stress affects the opening and closing of stomata by stimulating plant hormone signaling, and alleviates the water emission of stomata to improve the drought tolerance of plants. These results suggested that ABA-mediated pathways in stomatal closure may also play an important role in the regulation of water stress. Based on this, we can speculate that PP2C, SnRK2, ABF genes and DELLA proteins might have played a role in regulating stomatal closure, affecting the photosynthetic rate, and conserving water.

The generation of ROS is mainly due to the damage of the photosynthetic system in the aerial part and the effect of stomatal closure (Dietz, Turkan & Krieger-Liszkay, 2016). When plants are subjected to stressed, the ABA response accumulates and the ROS acts as a second messenger for ABA, the MAPK signal transduction pathway converts stress signals from a receptor into downstream response molecules that enhance drought resistance by modulating gene expression (Danquah et al., 2014). Previous studies demonstrated that ROS was an important messenger in the MAPK signal transduction pathway. Drought stress causes the accumulation of cellular ROS, which acts as a messenger molecule and damages membrane lipids (Jalmi & Sinha, 2015). MAPK signaling in the ABA-mediated MAPK signaling pathway synthesizing H2O2 is influenced by redox and ROS, thereby regulating intracellular ROS homeostasis and redox. Rentel and Liu have demonstrated that H2O2 generation stimulated the expression of the Arabidopsis OXI1 and results in high levels of H2O2 (Rentel et al., 2004; Liu et al., 2007). MAPK balances the production of reactive oxygen species through cascade, ROS acts as the second messenger of ABA signaling pathway and MAPK signaling pathway participates in ABA signaling pathway regulating stomata to adapt to drought stress (Matsuoka et al., 2018). Other studies have shown that the cascade of MAPK induction by ABA regulates plant growth and development to improve plant drought tolerance. Liu et al. (2022a); Liu et al. (2022b) have shown ABA, ROS, and calcium ion (Ca2+), played pivotal roles in controlling stomatal closure under drought conditions. In other words, H2O2 was likely responsible for inducing RbohD overexpression, and OXI1 act as a response element for H2O2 released from the respiratory burst, causing the expression of related genes by eliminating H2O2 to maintain intracellular homeostasis (Dietz, Turkan & Krieger-Liszkay, 2016). In this study, the KEGG pathway enrichment analysis revealed that 206 DEGs were significantly enriched, and the down-regulated DEGs in the roots of I. bungeana were significantly enriched in the MAPK signaling pathway. Seven respiratory burst oxidase RbohD genes and five OXI1 were enriched in roots. Three CAT1 were up-regulated in leaves tissues in the drought stress pathway Fig. 9. This was agree with previous findings that CT and H2O2 contents were significantly increased in leaves. Therefore, I. bungeana may induce the expression of RbohD, CAT, and OXI1 genes to counter the ROS generated during drought stress by activating the MAPK signaling pathway. ABA regulates ROS by inducing MAPK, and ROS acts as a second messenger in the ABA signaling pathway, coregulated stomatal size regulates plant growth.

IAA is primarily dependent on the tryptophan pathway. Early auxin-response genes include GH3 and SAUR (Luo, Zhou & Zhang, 2018). IAA, GH3 and SAUR are being expressed in response to auxin, thereby regulating the root system and plant height growth rates in plants to alleviate water deficiency (Meng, 2018). In this study, while most of the Auxin metabolism-related genes were downregulated in both tissues, involved twelve SAUR-related genes and eleven GH3-related genes. Nine ADP-ribosylation factor (ARF) were down-regulated in roots, eight ARF genes were down-regulated in leaves. However, AUX28 was the opposite (Fig. 11). These results suggest that the auxin response factor (ARF) may bind to the transcriptional repressor Aux/IAA and repress auxin synthesis thereby inhibiting the accumulation of biomass to alleviate water deficiency. Further, the root length, surface area, volume, forks number, and root biomass of I. bungeana were found to be significantly reduced. Furthermore, these genes IAA, GH3, SnRK2, ABF should be investigated further in I. bungeana.

Plants maintain normal plant water requirements by promoting starch hydrolysis to increase Ss content and maintain cellular osmotic pressure under drought stress (Alexou, 2013). α-amylase are enzymes that degrade starch and starch substrates (Janecek, Svensson & MacGregor, 2013), SUS, a key enzyme in the photosynthetic pathway, is involved in carbon resource allocation and the initiation of sugar signaling (Stein & Granot, 2019). Sucrose can be hydrolyzed by INV to produce glucose and fructose or reversibly converted to fructose and UDP-glucose via SUS hydrolysis. Drought stress increases soluble sugar content, activities of sucrose phosphate synthase, sucrose synthase, and acid invertase, and up-regulated the expression levels of GmSPS1, GmSuSy2, and GmA-INV, but decreases leaf Sc of soybean R2-R6 (Aliche et al., 2020). In this study, the SS content in leaves of I. bungeana was significantly increased compared to CK, whereas the starch content significantly decreased (p < 0.05) (Fig. 2F & Fig. 2L). SUS and beta-glucosidase were oppositely expressed in leaves and roots, which may be related to the enhanced carbohydrate metabolism activity under drought stress (Fig. 11). Additionally, the expression level of sucrose transporters SUC family were up-regulated in I. bungeana. Thus, the changes in sugar allocation, metabolism and transport in leaves could regulate the biomass allocation and photosynthesis of I. bungeana under drought stress. Glycolysis/Gluconeogenesis, with substantial up-regulation of GALM and PDC in the roots, was essential for enhancing the formation of alpha-D-Glucose-1P, an intermediate product in the starch and sucrose metabolic pathway (Fig. 11). Based on these findings, it was hypothesized that I. bungeana may up-regulate the glycolytic/glyco-isomeric pathway in roots to transport energy to above-ground parts through the root system, whereas leaves effectively resist water stress through sucrose regulation, and transferring energy from the source organ to the subsurface.

Conclusions

This experiment was conducted to study the effects on the growth of I. bungeana under water stress. Under drought conditions, I. bungeana enhances its survival ability and yield through enhance the antioxidant capacity, osmotic activities, withstand dehydration ability. Analysis by KEGG revealed that the genes responsible for plant hormone signaling, MAPK signaling pathway, as well as starch and sucrose metabolism were significantly enriched in leaves and roots. The higher level of transcription factors bHLH, bZIP, and MYB were majorly to activate downstream genes in significantly enriched pathway, which may mainly regulate stomatal movement to contributed to drought tolerance. The PP2C, SnRK2, ABF genes and DELLA proteins in response to ABA hormone might have played a role in regulating stomatal closure. From the metabolic pathway, it could be inferred that IAA, AUX28, SAUR, GH3 were down-regulated in roots response to accumulation of auxin in plant hormone signaling to reduce root structural characteristics. In the starch and sucrose metabolism pathway, SUS genes were significantly up-regulated in leaves, and sucrose transporter SUC was up-regulated in both leaves and roots. Therefore, we opined that the regulation of stomatal motility could be an important factor affecting the drought resistance of I. bungeana. Meanwhile, bHLH, bZIP, MYB, PP2C, SnRK2, ABF play an important role in the regulation of stomata responds to drought, and their roles in drought resistance in plant need to be further investigated. These findings, which may be valuable for drought resistance breeding, and to clarify the drought stress regulatory mechanisms of I. bungeana and other plants.

Supplemental Information

Supplemental Information 1 Raw data of morphological indicators and biomass for Table 2

Sheet: It represents an indicator. The ordinate is the indicator; the ordinate is the sample name in sheet

Click here for additional data file.

Supplemental Information 2 Raw data for the physiological indicators for Fig. 2

Sheet: It represents an indicator. The ordinate is the indicator; the ordinate is the sample name in sheet

Click here for additional data file.

Supplemental Information 3 Data on the number of transcription factors for Fig. 6

Abscissa: name; ordinate: data

Click here for additional data file.

Supplemental Information 4 The data of KEGG enrichement pathway for Fig. 7

Abscissa: name; ordinate: metabolic pathway

Click here for additional data file.

Supplemental Information 5 Raw data for Fig. 11

Abscissa: log2FC value in leaf and root; ordinate: gene serial number

Click here for additional data file.

Supplemental Information 6 Raw data for qPCR and RNA-seq for Fig. 12

Abscissa: name; Ordinate: data

Click here for additional data file.

Supplemental Information 7 The RNASeq Power grouped by RNA sequencing

Click here for additional data file.

We would like to thank D.r Shuxia LI, Jianqiang DENG and Qiaoli MA from Ningxia University for their helpful suggestions.

Additional Information and Declarations

Competing Interests

Author Contributions

Data Availability

The authors declare there are no competing interests.

Shuang Ma conceived and designed the experiments, authored or reviewed drafts of the article, and approved the final draft.

Haiying Hu conceived and designed the experiments, authored or reviewed drafts of the article, and approved the final draft.

Hao Zhang performed the experiments, prepared figures and/or tables, and approved the final draft.

Fenghua Ma performed the experiments, prepared figures and/or tables, and approved the final draft.

Zhihao Gao analyzed the data, prepared figures and/or tables, and approved the final draft.

Xueying Li analyzed the data, prepared figures and/or tables, and approved the final draft.

The following information was supplied regarding data availability:

The raw measurements are available in the Supplementary Files.

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
