# Peer review of "Physiological response and transcriptome analyses of leguminous Indigofera bungeana Walp. to drought stress"

_PeerJ, doi:10.7717/peerj.15440_

## Round 0.1 · original submission · Major Revisions

Dear Authors

The manuscript cannot be accepted for publication in its current form. It needs a major revision to be reconsidered for publication. The authors are invited to revise the paper considering all the suggestions made by the reviewers. Please note that requested changes are required for publication.

With Thanks

Reviewer 1 ·

Basic reporting

The paper entitled "Identiûcation and characterization of regulatory pathways associated with drought stress in leguminous Indigofera bungeana Walp. by transcriptome analysis” describes development of detailed transcriptome and analysis of DEG expression under drought stress in Indigofera bungeana Walp. The paper is very well written with detailed methods. The structure of paper is according to PeerJ standad.
However, there are few discrepancies please check all those carefully once again. I consider all of them as minor changes
Tittle. The authors also measured the physiological parameters of drought stress in Indigofera bungeana Walp. But they did not mention it in paper title rather focused only on regulatory pathways. Authors need to change the title and should include some physiological description
Introductions need more detail on drought stress especially of Indigofera bungeana Walp
Regarding the information on the figures - it is of high quality and necessary for presenting the data. Figure 8 labeling seems blur and difficult to read.
The authors should link the supplementary data to main text where it is necessary.

Experimental design

No comments

Validity of the findings

Very good work, excellent results, needs more in-depth discussion not merely stating the generalized discussion. The authors should include the comparative discussion that how their results are unique than others.

Additional comments

This is just a beginning I hope they will continue to identify gene functions through more experiments

Reviewer 2 ·

Basic reporting

I have gone through the manuscript. The manuscript is well written. The information presented in the manuscript is clear, no ambiguity found in the text. Sufficient literature provided. Still there are some discrepancies found throughout the manuscript, few are listed here for corrections:
1. Objective of the research should include in the abstract.
2. A clear concluding remarks should include at the end of the abstract
3. Keywords may be different from the words used in the title of the manuscript
4. There are several typing mistake found in the text, example 'fromby' in line 82
5. English language is ok but there are several problems, like 'have been the among the' line 17
6. References should be checked for journal style and missing

Experimental design

Experimental design is clear and address all research questions. Methodology is presented properly with references.

Validity of the findings

The research is deals with noble issues which may help to develop drought resistant fodder species for livestock farming. All relevant data have been provided with proper discussion and data have been statistically analyzed with standard tools. All necessary Tables and Figures are provided with proper discussion. Manuscript may accepted with minor corrections.

·

Basic reporting

Background:
Line 20: remove ‘t’ from the word analyses
Introduction:
Spacing problem throughout the introduction section.
Line 60: Remove repeated reference
Line 70: Are there other forage crops with drought resistance to establish a background? Establish a relevance of Lespedeza bicolor and Amorpha fruticosa in this context.
Overall structural organization is good.

Materials & Methods:
Spacing problem throughout the Materials & Methods section such as Line 107.
Overall, grammar can be improved in this section such as Line 139.
Provide valid references throughout this section for each procedure such as Line 111.
Line 100: sentence structure needs to be corrected.
Provide full forms Line 117, where these terms are used at first in the article.
Correct the scientific symbols such as Line 118.
Try not to split the words in two lines without a hyphen such as Line 140.

Results:
Provide detailed legends for tables and figures so that they can be represent the respective story.
Again, major problem of spacing, split words, abbreviations throughout the section.
Figure 1. Correct caption grammar and provide more details in the legend about the differences.
Figure 2. Provide full forms for all terms in the legend. Add alpha level of significance in all the sub-figures. What does all the axes represent?
Figure 3. Provide details about the matrix, scale, and findings. What does different colors represent?
Table 3. Provide full forms for all groups in the legend.
Figure 4a. Add exact number of genes up/down regulated in the respective bars.
Figure 4b & c. Same legends for both the figures? What different information do they provide?
Figure 5. Remove repetition in legend.
Figure 8. Provide clear and detailed explanation in text section.

Supplementary information:
Provide an introductory sheet to explain what kind of data is in supplementary section and to which main figure/table/text it belongs.
What are in the graphical files? Are these supporting any main findings and needed? Files does not open.

Experimental design

Line 130: Provide more information in detail about RNA extraction: which part, replications, developmental stage etc.
Line 146: Establish a detailed experimental design information may be a table?

Validity of the findings

Results: Figure quality is not such as Figure 8 – most of the text is not readable.
Explain GO figure 5 to provide a context of differences between leaf and root better.
Figure 6. Provide more relevance in results section?
Figure 7. Provide a clear explanation for all the components in all sub-figures.
Figure 8. Poor quality, text not readable, repeatability in the legend. Explain findings.

Discussion:
Overall structure is good. However, a better flow of information on to connect different findings is needed.

Additional comments

Overall, the objective, methods and findings are relevant and provide a good scientific story. However, the overall structural framework is not great. There is a major problem of spacing, and grammatical errors throughout the manuscript. No scientific references are providing for most of the procedures in methods section. Visual quality of figures is not good. More efforts are needed for a better flow of results and discussion. Results section only includes larger captions of all the tables and figures. Provide a better context to the story. All the changes are doable to improve the manuscript quality.

Reviewer 4 ·

Basic reporting

The whole manuscript requires serious revision in punctuation and style. Especially references, whether in-text citations or the complete reference list. Besides being unorganized, I could not locate references due to the presence of single letters, the rest of the words, and unformatted lines in the reference list.
Subtitles should be bold.
The first paragraph in the introduction is very general and gives no specific information about your work.
The second paragraph would be enriched by previous transcriptome analysis attempts instead of focusing on specific genes, which would serve you better in the discussion part.
Please join the test material and experimental treatment in one paragraph with one subtitle.
In lines 146-7, what are those codes out of sudden, and why do authors use such codes without any prior description?
Indigofera should be abbreviated afterward; no need to write the full name after being mentioned once.
Results in numbers and details are missing; only conclusive sentences and outputs in general results should be stated clearly. For example line 181, what numbers indicate that stress significantly inhibits photosynthesis? Another example: line 191: different degrees of correlations existed, what are the numbers for each degree, should readers rely on the Figures and tables to find the information of your work?! Another conclusion is drawn in lines 267-268 within the results part, which is incorrect to be written there.
Table 1 is found cited in the text and in the supplementary files.
Some supplementary files are in a weird format; authors should be concise and transparent regarding proof of work or a supporting file for their outputs.
Figures require clarity; some need refining for better resolution others need better grouping and design. Some are too big, while others are dense with information and are given in low resolution. Please be consistent with the letter you use for the inner figures; letters have a different font, size, and color among all figures.
In detail: in Fig 1, all the black margins can be cropped. Fig 2, but for the graph letters on the middle top or right top of the graphs, use a typical graph prism histogram, which will help you order them better. Please reduce the height of the histograms, as they are stretched and cause low resolution. As you have one-way ANOVA with two groups only, please discard the a and b letters, and use an asterisk (*) to define the significance levels (*, **, or ***). Fig 3, as you didn’t write the correlation parameters, not the program you used, I suspect this figure is correct. Levels of correlation are essential unless they are significant. Please add an asterisk (*) to define the significance levels for each box, if any. Otherwise, change them to white color and indicate that they were not significant. Fig 4, please reorder the items and leave the venn shapes as default; don’t resize them or resize through the corner point to not stretch the figure incorrectly. Fig 5, I can’t see anything; please make it bigger using the corner point of each photo. Fig 6, TF is not equally written; one photo is marked with a, while the other is not; both figures are stretched height with no reason; please resize to be homogenized with the rest of the figures. For Fig 7, please check the comments on Fig 5. Figures 9 and 10 are blurred and not looking good; very oversized, unhomogenized, and undefined properly in the figure legend.

Please don't state sentences in discussion without citation unless they are your findings.

Experimental design

In lines 70-73. The study is based on a reference: Xu Weiwei, et al. (2017). However, I couldn’t locate it, which will weaken the justification for using this plant as a drought-tolerant model.
The experimental design is vague; it requires a clear strategy for maintaining moist and drought conditions. How you measured by weighing at a fixed time in the afternoon every day, how many grams you used, and what changes in total water content did you find after four weeks from a 2.5kg soil? How many samples did you cultivate, and how many did you apply your treatment on? Please rewrite and include more details for clarity.
What kind of reference you used to map your RNAseq reads? Why you used the unigenes mentioned in Figure 10?
How do you annotate the gene function? You only mention the databases, but the how is not mentioned.
Why have you used DESeq, not DESeq2? Any justification for that retroaction
GraphPad do both ANOVA and graphs; why you used Excel and SPSS? At what conditions did you set your parameters? You didn’t mention the program used for the person correlation! What kind of data transformation have you been using to compare different assays, knowing that each assay has a different type of numerical scale?
In figure 11, the qPCR in fold change is correlated to RNAseq reads, which is in FPKM or what? Those data types are different and should not be tested for correlation unless a proper data transformation is applied.

Validity of the findings

Novality is not assured unless the information given about the plant is approved by known sources of literature. Raw data is unavailable; most represented information can be widely found in other plants with no novel findings. The lack of precise details for the followed methodology raises some issues regarding the awareness of the authors of the RNAseq work protocol. I have difficulties following the narration of the work and the cited references. Authors should provide access to the raw data by depositing it in public databases.

---

## Round 0.2 · Minor Revisions

Dear Authors

The manuscript needs a minor revision to be reconsidered for publication. The authors are invited to revise the paper considering all the suggestions made by the reviewers. Please note that requested changes are required for publication.

With Thanks

Reviewer 2 ·

Basic reporting

Author revised his manuscript as per the comments made earlier.
1. added study objective in the abstract
2. added conclusion
3. keywords can change using words other than used in the title.
4. English is ok now

Experimental design

OK

Validity of the findings

Findings is ok, manuscript may be accepted for publication in Peer J

Additional comments

Not applicable

·

Basic reporting

Please check your spellings in the rebuttal letter. And make sure the line numbers are continuous so that the reviewers can track any changes the authors made. Overall, authors covered basic reporting section. However, the figures still need more work. Please cover all the previous comments.

Experimental design

Satisfied.

Validity of the findings

Please address the previous comments thoroughly.

Additional comments

NA

Reviewer 4 ·

Basic reporting

All the addressed issues have been considered by the authors.

Experimental design

No comment

Validity of the findings

No comment

---

## Round 0.3 · accepted · Accept

Dear Authors,

I am pleased to inform you that after the last round of revision, the manuscript has been improved a lot, and it can be accepted for publication.
Congratulations on accepting your manuscript, and thank you for your interest in submitting your work to PeerJ.

With Thanks

The Academic Editor noted that the resolution of Figure 1 is low so please work with the production staff to improve it.